# SBRT of Spinal Metastases Using a Simultaneous Integrated Boost Concept in Oligometastatic Cancer Patients Is Safe and Effective

**DOI:** 10.3390/cancers15245813

**Published:** 2023-12-12

**Authors:** Maria Waltenberger, Christian Strick, Marco M. E. Vogel, Christian Diehl, Stephanie E. Combs

**Affiliations:** 1Department of Radiation Oncology, Klinikum Rechts der Isar, Technical University of Munich (TUM), 81675 Munich, Germany; christian.strick@mri.tum.de (C.S.); marco.vogel@tum.de (M.M.E.V.); christian.diehl@mri.tum.de (C.D.); stephanie.combs@mri.tum.de (S.E.C.); 2German Cancer Consortium (DKTK), Partner Site Munich, a Partnership between DKFZ and University Hospital Klinikum Rechts der Isar, 81675 Munich, Germany; 3Institute of Radiation Medicine (IRM), Helmholtz Zentrum München, 85764 Neuherberg, Germany

**Keywords:** spinal metastases, SBRT, stereotactic body radiotherapy, SIB, dose escalation, VCF, oligometastatic

## Abstract

**Simple Summary:**

Metastases-directed stereotactic body radiotherapy (SBRT) is increasingly used in the context of oligometastatic disease, as fist prospective data show improved survival with additional ablative radiotherapy in subgroups of oligometastatic patients. The spine is a common site for metastases, and different treatment regimens are used for spinal SBRT. Vertebral compression fracture (VCF) is a main safety concern. Our single-center retrospective analysis of 62 consecutive oligometastatic cancer patients with 71 spinal metastases provides first evidence for spinal SBRT utilizing a simultaneous integrated boost to the macroscopic lesion in a larger patient cohort and demonstrates the safety and effectivity of this treatment regimen: No ≥ grade III acute and one grade III late toxicity (VCF) occurred, and estimated local control rates were excellent and durable with 98.6% and 96.4% at one and two years. Regarding survival, patients with prostate cancer, secondary oligometastatic disease and good performance status might especially benefit from this treatment approach.

**Abstract:**

(1) Purpose: To assess the safety and effectivity of stereotactic body radiotherapy (SBRT) on spinal metastases utilizing a simultaneous integrated boost (SIB) concept in oligometastatic cancer patients. (2) Methods: 62 consecutive patients with 71 spinal metastases received SIB–SBRT between 01/2013 and 09/2022 at our institution. We retrospectively analyzed toxicity, local tumor control (LC), and progression-free (PFS) and overall survival (OS) following SIB–SBRT and assessed possible influencing factors (Kaplan–Meier estimator, log-rank test and Cox proportional-hazards model). (3) Results: SIB–SBRT was delivered in five fractions, mostly with 25/40 Gy (*n* = 43; 60.56%) and 25/35 Gy (*n* = 19, 26.76%). Estimated rates of freedom from VCF were 96.1/90.4% at one/two years. VCF development was significantly associated with osteoporosis (*p* < 0.001). No ≥ grade III acute and one grade III late toxicity (VCF) were observed. Estimated LC rates at one/two years were 98.6/96.4%, and histology was significantly associated with local treatment failure (*p* = 0.039). Median PFS/OS was 10 months (95% CI 6.01–13.99)/not reached. Development of metastases ≥ one year after initial diagnosis and Karnofsky Performance Score ≥ 90% were predictors for superior PFS (*p* = 0.038) and OS (*p* = 0.012), respectively. (4) Conclusion: Spinal SIB–SBRT yields low toxicity and excellent LC. It may be utilized in selected oligometastatic patients to improve prognosis. To the best of our knowledge, we provide the first clinical data on the toxicity and effectivity of SIB–SBRT in spinal metastases in a larger patient cohort.

## 1. Introduction

The spine is a common site for metastases with up to 50% of all cancer patients developing spinal metastases during the course of their disease [1]. Spinal metastases pose a risk for relevant morbidity such as severe pain, fracture or spinal cord compression [2], impairing overall quality of life [3] and requiring radiotherapy or surgical intervention.

Radiotherapy is the standard-of-care treatment for painful spinal metastases that do not require decompression or stabilization surgery, and is mostly and historically delivered utilizing palliative radiation doses [4]. There are a variety of recently conducted clinical trials investigating stereotactic body radiotherapy (SBRT) for pain relief in spinal metastases [5,6,7]. As the results are inconclusive, they do not justify the routine use of SBRT for painful spinal metastases. Van der Velden et al. recently published an ASTRO ACROP guideline on radiotherapy in the treatment of uncomplicated spinal metastases [8] recommending to use conventionally fractionated radiotherapy as clear evidence for the superiority of SBRT for pain control is currently lacking. 

SBRT is however increasingly used in the context of oligometastatic disease including spinal lesions, yielding promising results with improvements in survival [9,10,11]. Oligometastatic disease, first introduced by Hellmann and Weichselbaum in 1995 [12], is defined as a subcategory of tumor spread, as an intermediate state between localized disease and advanced metastatic cancer. There is no uniform definition regarding the number of metastases that are considered oligometastatic; however, oligometastatic disease is usually regarded as being equivalent to a maximum of three to five metastases [9,13,14]. Recently, a detailed ESTRO EORTC guideline for the characterization and classification of oligometastatic disease has been published [15].

A variety of dose schemes that are applied for spinal SBRT exist, ranging from a single-fraction regimen [9,16] to as many as ten fractions [17]. We recently conducted a survey on treatment concepts for spinal metastases among radiation oncologists of the German Society for Radiation Oncology and found spine SBRT practice to be considerably heterogenous [18]. However, we identified two dose schemes that were most frequently used among participants, i.e., 27 Gy in three fractions and 30 Gy in five fractions. Consensus guidelines on target volume definition for spinal SBRT are available both for the mobile spine [19] and the sacrum [20]. It is recommended that treatment volumes include a bony margin beyond the macroscopic tumor enclosing abnormal bone marrow signal suspicious for micrometastases and regular bone marrow as well to account for subclinical spread. The extension of the target volume also depends on the location of the metastasis within the vertebra. Chen et al. recently demonstrated that adherence to consensus guidelines for target volume definition results in improved local tumor control [21].

SBRT in spinal metastases yields good and durable local tumor control. For instance, Chen et al. report local control rates of 81.1% and 70.6% at one and two years following SBRT with cumulative doses of 24 to 27 Gy in two to three fractions [21]. Guckenberger et al., for instance, observed an 82% local tumor control rate at a median follow-up period of 45 months after SBRT with 5 × 7 Gy or 10 × 4.85 Gy [22], and Ning et al. report excellent 5-year local control rates of 91% after one- and three-fraction SBRT with cumulative doses of 24 and 27 Gy, respectively [23]. 

SBRT-related side effects are considerably low; however, vertebral compression fracture (VCF) and pain at the treated site in particular do occur. Neurotoxicity such as radiculopathy or myelopathy is rare. VCF rates up to 39% after SBRT are reported [24], with the majority of patients usually not requiring surgical intervention [22,23]. However, salvage surgery rates after SBRT-related VCF of up to 45% have been observed [25]. VCF often develops shortly after treatment, but can occur years beyond SBRT as well [22,23]. Several risk factors for VCF development are known. Risk can be estimated with the Spinal Instability Neoplastic Score (SINS) [26], a score originally developed to select patients that may benefit from surgical intervention and composed of the items localization, pain, bone lesion quality, spine deformity, vertebral body collapse and posterolateral involvement. A Bilsky Score >0, i.e., epidural extension, has been described as being associated with VCF development after spine SBRT as well [27].

In the following, we report our single-institution experience with fractionated spinal SBRT using a simultaneous integrated boost (SIB) concept (SIB–SBRT) in oligometastatic cancer patients and investigate the safety and effectivity of this treatment approach. 

## 2. Materials and Methods

The institute’s own database was screened for patients having received SIB–SBRT delivered with a linear accelerator between January 2013 and September 2022, allowing for a follow-up period of at least 12 months. Patients who were not oligometastatic, i.e., had more than five metastases present at the timepoint of SIB–SBRT, were excluded from the analyses. A total of 62 consecutive oligometastatic patients with 71 spinal metastases were identified. Data were collected retrospectively and data lock was on 7 October 2023. A vote was obtained from the local ethical committee (Technical University of Munich) prior to conducting the study (2019-476_1-S-SR). 

Toxicity was assessed according to CTCAE c4.03 [28]. VCF rates, local control (LC), progression-free survival (PFS) and overall survival (OS) were assessed with Kaplan–Meier estimator. Time intervals were calculated from the last SIB–SBRT fraction to VCF, (local) progression or death. For VCF rates and LC, patients without local event were censored at the timepoint of the last available local imaging. For PFS, patients without evidence of progressive disease were censored at the timepoint of the last available staging imaging and for OS, patients still alive or lost to follow-up were censored at the timepoint of the last contact with our institution. 

Parameters possibly influencing VCF, LC, PFS and OS were tested for significance as follows. Categorial variables (gender, primary tumor, Karnofsky Performance Score (KPS), postmenopausal status, osteoporosis, obesity, oligometastatic subgroup, number of active metastases present at SIB–SBRT, location of spinal metastasis, type of metastasis, previous local therapy, baseline pain, SINS, treatment planning imaging, SIB–SBRT dose and volumes and systemic therapy) were summarized and tested for significance in their respective categories utilizing the log-rank test. For KPS, patients with no or minimal symptoms (KPS 90-100) were compared to patients with some to considerable symptoms (KPS 80 and worse) regarding survival (log-rank test). Continuous variables (age at primary diagnosis, at the development of metastases and at SIB–SBRT as well as time between initial diagnosis and metastases development) were summarized in age categories or time intervals and with median values, and were tested for significance with a Cox proportional-hazards model as continuous parameters and additionally with the log-rank test, dichotomously dividing the study population at the median value of the respective parameter. 

The significance level was set to α = 0.05 (two-sided). All parameters considered significant in univariate analyses (*p* < 0.05) were included in a subsequent multivariate analysis using a Cox proportional-hazards model. For the log-rank test *p*-values, and for the Cox proportional-hazards model *p*-values and the corresponding hazard rations (HR) and 95% confidence intervals (95% CI), are reported. Statistical analyses were carried out with SPSS version 27 (IBM, Armonk, NY, USA).

## 3. Results

### 3.1. Patient Characteristics

Patient characteristics are shown in Table 1. Approximately two thirds (*n* = 43, 69.35%) of patients were male and 72.09% of them had prostate cancer (*n* = 31). Primary diagnoses in other patients were breast (*n* = 9, 14.52%), lung (*n* = 7, 11.29%), head and neck cancer (*n* = 4, 6.45%) and melanoma (*n* = 3, 4.84%). In the remaining 12.90% of patients (*n* = 8), summarized in the category “other”, primary diagnosis was cholangiocarcinoma, leiomyosarcoma, cervical cancer, Merkel cell carcinoma, pancreatic neuroendocrine tumor, rhabdomyosarcoma, paraganglioma and adenocarcinoma of the stomach in one case each. The median age at initial diagnosis was 60.5 years (range (R) 20–91) and the median time between initial diagnosis and SIB–SBRT was 28.5 months (R 0.5–337).

The majority of patients (*n* = 48, 77.42%) had a KPS of 90% or higher at the timepoint of SIB–SBRT. Seven patients (11.29%) received simultaneous spinal SIB–SBRT to two lesions and one patient (1.61%) to three lesions; in all other cases (*n* = 54, 87.10%), a single spinal metastasis was treated with SIB–SBRT. Thoracic (*n* = 34, 47.89%) and lumbar (*n* = 30, 42.25%) spine were the most common locations, and metastases were mostly osteoblastic (*n* = 45, 63.38%) and asymptomatic (*n* = 62, 87.32%). Nine lesions (12.68%) were painful prior to SIB–SBRT, at a median intensity of four out of ten (R 2–10) according to the numeric rating scale (NRS). All patients with pain of 5/10 NRS or higher used non-opioid analgesics for pain control. Classified according to the SINS, 94.37% (*n* = 67) of lesions were stable and 5.63% (*n* = 4) were potentially unstable.

Staging imaging was in all cases carried out prior to SIB–SBRT. Thirty-one patients (50.00%) had secondary oligometastatic disease (no metastases at initial diagnosis) and 10 patients (16.13%) were diagnosed with primary oligometastatic disease. The other remaining patients previously had extended metastatic disease and were classified as oligoprogressive (*n* = 19, 30.65%) or oligopersistent (*n* = 2, 3.23%). In the majority of cases (*n* = 39, 62.90%), only a single active metastasis was present at the timepoint of SIB–SBRT, and the maximum number was three. If further metastases were present, local ablative treatment with SBRT or resection was performed in all cases but one. This one patient presented with secondary oligometastatic melanoma with two bone and one soft tissue metastasis on the deltoid muscle. Ablative therapy could not be offered for the soft tissue metastasis but immunotherapy was initiated. Systemic therapy was part of the treatment concept for oligometastatic disease in 40 cases (64.52%), either chemotherapy, immunotherapy or anti-hormonal therapy, or a combination. Nine patients (14.52%) additionally received antiresorptive agents, i.e., Denosumab or zoledronic acid. Twenty-two patients (35.48%) were treated with local ablative therapy only. Two lesions (2.82%) received surgery prior to SIB–SBRT.

### 3.2. SBRT Setup, Target Volume Definition and Dose Prescription

Patients were treated with a linear accelerator equipped for SBRT (Varian Medical Systems, Palo Alto, CA, USA) utilizing a 6 or 15 MeV photon beam. Treatment planning computed tomography (CT) was acquired in supine position and with 3 mm slice thickness. The immobilization method was generally dependent on the height of the metastatic lesion with stereotactic masks (Brainlab, Munich, Germany) being used for lesions in the cervical and upper thoracic spine, and a vacuum fixation cushion (BlueBAG™, Elekta, Stockholm, Sweden) for metastases in the middle thoracic, lumbar and sacral spine. Knee support was utilized for all patients and abdominal press (*n* = 8, 12.90%), foot support (*n* = 9, 14.52%) and 4D-CT (*n* = 12, 19.35%) were occasionally used. 

Treatment planning magnetic resonance imaging (MRI) and positron emission tomography (PET) were available in 67 (94.37%) and 39 (54.93%) cases, respectively. In three of the four cases without treatment planning MRI, PET imaging was obtained. All lesions were treated in five fractions and the dose was prescribed to the median (100% of the dose covers 50% of the target structure) in the majority of cases (*n* = 57, 80.28%). Other dose prescription approaches were as follows: 80% of the dose covers 96–97% of the target structure (*n* = 2, 2.82%), 95% of the dose covers 75–100% of the target structure (*n* = 8, 11.27%) and 100% of the dose covers 55–95% of the target structure (*n* = 4, 5.63%) with target structure generally referring to SIB. 

The most frequently applied dose scheme (*n* = 43, 60.56%) was 5 × 5/8 Gy, followed by 5 × 5/7 Gy (*n* = 19, 26.76%) and 5 × 6/8 Gy (*n* = 5, 7.04%). SIB–SBRT was delivered with 5 × 5/6 Gy in two cases and with 5 × 5.5/8 Gy and 5 × 4/8 Gy in one case each. The median BED_10_ prescription doses were 37.5 Gy (R 28–48) for PTV and 72 Gy (R 48–72) for SIB volumes, respectively, and median BED_3_ prescription doses for prostate cancer metastases were 66.67 Gy (46.67–90) for PTV and 146.67 Gy (90–146.67) for SIB. VMAT (RapidArc^®^, Varian Medical Systems, Palo Alto, USA) technique was used in the vast majority (*n* = 69, 97.18%) and stereotactic 3D technique with eight and ten different fields in two cases (2.82%).

Planning target volume (PTV) definition was essentially and mostly based on the aforementioned International Spine Radiosurgery Consortium consensus guidelines for target volume definition in spinal stereotactic radiosurgery by Cox et al. and on the international consensus recommendations for target volume delineation specific to sacral metastases and spinal stereotactic body radiation therapy (SBRT) by Dunne et al. As recommended by the consensus guidelines, compartment(s) where the metastasis was located and, if applicable, the adjacent compartment(s) were generally included in the PTV. Additionally, a simultaneous integrated boost (SIB) was delivered to the macroscopic metastasis. Figure 1 shows exemplary treatment plan excerpts and Table A1 (Appendix A) summarizes the strategies for PTV definition depending on the localization of the metastasis in the vertebra at patient level. Median PTV and SIB volumes were 74.9 (R 10.9–352.2) and 7.1 mL (R 0.7–202.4), respectively.

### 3.3. Treatment Toxicity

No acute ≥ grade III side effects were observed. One patient each (1.61%) suffered from grade I diarrhea, dysphagia and radiodermatitis during SIB–SBRT. Sixteen patients (25.81%) reported grade I fatigue, thirteen of which received systemic therapy immediately prior or concomitant to SIB–SBRT. Grade I nausea was reported by five patients (8.06%) and grade II nausea by one patient (1.61%). Pain during SIB–SBRT was reported by 15 patients (24.19%) with a total of 16 lesions (22.54%), at a median intensity of 3/10 NRS (R 2–6/10). Nine of those patients had to use non-opioid analgesics. Pain flare during SIB–SBRT, i.e., the development of de novo pain, was observed in eight patients (12.90%) with nine lesions (12.68%), at a median intensity of 3/10 NRS (R 2–4/10). The other seven patients (11.29%) with seven metastases (9.86%) already had painful lesions before treatment, and pain was stable during SIB–SBRT in three cases, improved by a median of 2 points on NRS (R1–7) in three cases and worsened by 1 point on NRS in one patient. Complete pain relief during SIB–SBRT was seen in two patients. Forty-five patients (72.58%) with 52 lesions (73.23%) were free of pain before and during SIB–SBRT, while one additional patient had one painful and one pain-free lesion. Figure 2 illustrates pain dynamics at the individual patient level.

Clinical information on late side effects (i.e., systematic radiooncological follow-up of at least three months after treatment) was available for 46 patients with 52 spinal metastases, with a median clinical follow-up time of 15.5 months (R 3–113). Skin fibrosis or hypo-/hyperpigmentation were not observed, but grade I dry skin was seen in one patient (2.17%). Symptomatic myelopathy or radiculopathy were not observed. Acute nausea ultimately resolved in all cases with available long-term follow-up (*n* = 5). Fatigue during SIB–SBRT completely resolved in 50% of patients with long-term follow-up (*n* = 6) and was stable at grade I in the remaining patients (*n* = 6). All but one patient with stable grade I fatigue received systemic therapy after SIB–SBRT. There were six patients (6.52%) with six lesions (11.54%) who reported pain three months or more after SIB–SBRT at a median intensity of 3.5/10 NRS (R1–8/10; see Figure 2), requiring non-opioid and opioid analgesics in two cases each. Four patients had VCF-associated pain. Pain ultimately completely resolved in two of these cases after intervention for VCF, resulting in a proportion of 8.70% of patients and 7.69% of lesions with pain at the end of follow-up. Thirty patients (65.22%) with 36 lesions (69.23%) were free of pain before, during and after SIB–SBRT. 

Local follow-up imaging was available for *n* = 57 patients with *n* = 65 spinal metastases, with a median local follow-up time of 18 months (R 1–112). There were four SIB–SBRT-related VCF events at the treated site (6.15%) that developed 3, 4, 5 and 9 months after therapy and one case where VCF developed 4 months after therapy due to tumor progression (see below). Excluding the one patient with progression-related VCF, estimated rates of freedom from VCF after SIB–SBRT were 96.1% (SE 0.03) and 90.4% (SE 0.05) at one and two years, respectively (see Figure 3a). *n* = 21 lesions (32.81%) were lost to follow-up during the first year after SIB–SBRT, in five cases (five patients, 7.81%) due to death; and *n* = 44 lesions (68.75%) were lost to follow-up during the first year after SIB–SBRT, in nine cases (eight patients, 14.06%) due to death. Considering only lesions with local follow-up imaging available at one (*n* = 45) and two years (*n* = 23) after SIB–SBRT results in observed VCF rates of 8.89% at one and 17.39% at two years after treatment. Two patients had VCF-associated pain that was self-limiting, requiring opioid analgesics in one case. Two patients required surgical intervention for VCF, including the one patient with tumor-associated VCF. PTV extended well beyond the vertebral body in all cases where VCF developed after SIB–SBRT and included the whole vertebra in two cases. VCF in an adjacent level was seen in two cases (3.08%), one asymptomatic and the other associated with minimal self-limiting pain that did not require analgesics. Risk for VCF development at the treated site after SIB–SBRT was significantly associated with osteoporosis (*p* = 4.36 × 10^−14^). Neither total SINS (*p* = 0.51) nor its individual components (see Appendix A Table A2) were associated with freedom from VCF. Likewise, no significant correlation was found for any of the following parameters: gender (*p* = 0.68), histology (*p* = 0.10), obesity (*p* = 0.89), previous local therapy at the treated site (*p* = 0.73), use of systemic, anti-hormonal or antiresorptive therapy in the treatment concept for oligometastatic disease (*p* = 0.49, *p* = 0.87 and *p* = 0.24, respectively), postmenopausal status (*p* = 0.57), location (*p* = 0.58), SIB dose (*p* = 0.94) and volume (*p* = 0.30). There was a trend for higher risk of VCF with larger PTV volumes (*p* = 0.08). Since only osteoporosis was significant in univariate analyses, multivariate analysis was not carried out.

### 3.4. Local Tumor Control

Imaging for local control assessment was available for *n* = 57 patients with *n* = 65 lesions (see above). All patients without local follow-up imaging (*n* = 5 patients with *n* = 6 lesions) had prostate cancer and received regular controls of prostate-specific antigen (PSA), with a median time from SIB–SBRT to last PSA control of 13 months (R 1–36). A PSA drop was observed in all of the five patients after SIB–SBRT and there was no case of PSA level increase afterwards. Antiandrogen therapy status in these five patients was as follows: Two patients never received antiandrogen treatment and were treated with SIB–SBRT only for metastatic disease, one patient did receive antiandrogen therapy as part of the initial therapy for localized prostate cancer but not in the metastasized context, and two patients started antiandrogen treatment shortly before SIB–SBRT when the respective spinal metastases were detected and continued treatment afterwards. Based on the PSA history after SIB–SBRT, we assumed local tumor control in these five patients. Patients with prostate cancer generally received re-staging with (prostate-specific membrane antigen) PSMA-PET-CT in case of PSA level elevation after SIB–SBRT. Median time to last local follow-up imaging or (if none available) PSA was 18 months (R 1–112). 

Two cases (2.82%) of local progression were detected, resulting in estimated local control rates of 98.6% (SE 0.01) and 96.4% (SE 0.03) at one and two years after SIB–SBRT (see Figure 3b). *n* = 25 lesions (35.21%) were lost to follow-up during the first year after SIB–SBRT, in five cases (five patients, 7.04%) due to death; and *n* = 50 lesions (70.42%) were lost to follow-up during the first year after SIB–SBRT, in ten cases (nine patients, 14.08%) due to death. Considering only lesions with local follow-up available at one (*n* = 47) and two years (*n* = 21) after SIB–SBRT results in observed local control rates of 2.13% at one year and 9.53% at two years after treatment. Both local recurrences were in-field. One patient who suffered from secondary oligometastatic cervical cancer and presented with a bifocal metastasis in one vertebral height treated with 5 × 5/8 Gy had local progression at one month after SIB–SBRT. Both macroscopic lesions (i.e., SIB volumes) were progressive. In the context of local tumor progression, the patient developed a VCF and had to undergo stabilization surgery. At the time of local recurrence, a single new liver metastasis was detected as well. The patient received SBRT to the liver metastasis and immunotherapy was initiated. A second patient with primary oligometastatic Merkel cell carcinoma and a single spinal metastasis treated with 5 × 5/8 Gy showed a lesion suspicious for local recurrence 14 months after SIB–SBRT. The lesion was located within the PTV but outside the SIB volume. This was the only lesion suspicious for vital tumor in this elderly patient (87 years of age) in reduced general condition, and therefore the well-tolerated immunotherapy was continued. Re-staging was scheduled at a three-month interval (pending at the timepoint of this analysis).

Histology had a significant impact on local tumor control (*p* = 0.039), showing highly significant differences between patients with prostate cancer and “other” (see patient characteristics, *p* = 0.007). Neither previous local therapy (*p* = 0.82), nor use of systemic therapy in the treatment concept for oligometastatic disease (*p* = 0.62), treatment planning imaging (MRI and PET available yes/no *p* = 0.77 and *p* = 0.89, respectively), target volume (PTV and SIB volume *p* = 0.14 and *p* = 0.95, respectively) or dose (PTV and SIB dose *p* = 0.97 and *p* = 0.87, respectively) were significantly associated with local tumor control. Since only histology was significant in univariate analyses, multivariate analysis was not carried out.

### 3.5. Progression-Free Survival after SIB–SBRT

Re-staging imaging was available for 53 patients (85.48%) and all patients without follow-up imaging had prostate cancer and PSA follow-up (*n* = 9, 14.52%). Progression was either stated when re-staging imaging showed progressive disease, or—for patients with prostate cancer and no follow-up imaging—in case of increasing PSA levels after SIB–SBRT. Antiandrogen therapy status in patients with prostate cancer and no follow-up imaging was as follows. Three patients never received antiandrogen treatment and were treated with SIB–SBRT only for metastatic disease, one patient did receive antiandrogen therapy as part of the initial therapy for localized prostate cancer but not in the metastasized context, and five patients received antiandrogen treatment when the respective spinal metastases were detected and continued treatment after SIB–SBRT. Median time from treatment to last follow-up imaging or last PSA value was 18 months (R 1–83), and 46 patients (74.19%) had progressive disease. Median PFS was 10 months (95% CI 6.01–13.99, see Figure 4a). 

There were no significant differences in PFS after SIB–SBRT by tumor type (*p* = 0.11). Patients with prostate cancer, however, had a significantly longer PFS than patients with lung cancer (*p* = 0.003, see Figure 5a). Patients that developed metastases ≥ one year after initial diagnosis had a significantly longer time to progression than patients who developed metastases earlier during the course of their disease (*p* = 0.000826, see Figure 5b), but there were no significant differences for time between initial diagnosis and development of metastases tested as a continuous variable (*p* = 0.057, HR 0.992, 95% CI 0.98–1). We consistently observed that subclassification of oligometastatic state had a significant impact on PFS as well (*p* = 0.014), showing longer PFS rates for patients with secondary oligometastatic disease compared to primary oligometastatic disease (*p* = 0.004) and oligoprogression (*p* = 0.008, see Figure 5c). Gender (*p* = 0.18), age at initial diagnosis (*p* = 0.72 for log-rank test and *p* = 0.71, HR 1.004, 95% CI 0.98–1.03 for Cox regression), age at the development of metastases (*p* = 0.22 for log-rank test and *p* = 0.79, HR 0.997, 95% CI 0.98–1.02 for Cox regression), age at SIB–SBRT (*p* = 0.18 for log-rank test and *p* = 0.60, HR 0.995, 95% CI 0.97–1.02 for Cox regression), number of metastases at SIB–SBRT (*p* = 0.63), use of systemic therapy in the treatment concept for oligometastatic disease (*p* = 0.41), KPS (*p* = 0.62) and dose (PTV and SIB dose *p* = 0.89 and *p* = 0.21, respectively) did not significantly impact PFS. In multivariate analysis, only time between initial diagnosis and development of metastases remained significant (*p* = 0.038).

### 3.6. Overall Survival after SIB–SBRT

During the observation period, 12 patients (19.35%) died. Median OS from SIB–SBRT until death was not reached. Estimated survival rates at one, two and five years after SIB–SBRT were 91.5% (SE 0.04), 80.2% (SE 0.06) and 64.8% (SE 0.10), respectively (see Figure 4b). 

There were no significant differences in OS for histology (*p* = 0.11), but subgroup analyses showed that patients with prostate cancer lived significantly longer after SIB–SBRT than did patients with lung (*p* = 0.014) and head and neck cancer (*p* = 0.001, see Figure 6a). A KPS ≥ 90% at SIB–SBRT was significantly associated with longer OS (*p* = 0.012, see Figure 6b). Gender (*p* = 0.18), age at initial diagnosis (*p* = 0.42 for log-rank test and *p* = 0.51, HR 0.987, 95% CI 0.95–1.03 for Cox regression), age at the development of metastases (*p* = 0.09 for log-rank test and *p* = 0.62, HR 0.99, 95% CI 0.95–1.03 for Cox regression), age at SIB–SBRT (*p* = 0.20 for log-rank test and *p* = 0.50, HR 0.985, 95% CI 0.94–1.03 for Cox regression), time between initial diagnosis and the development of metastases (*p* = 0.84 for log-rank test and *p* = 0.61, HR 1003, 95% CI 0.99–1.01 for Cox regression), number of metastases at SIB–SBRT (*p* = 0.74), use of systemic therapy in the treatment concept for oligometastatic disease (*p* = 0.67), subclassification of oligometastatic state (*p* = 0.21) and dose (PTV and SIB dose *p* = 0.53 and *p* = 0.21, respectively) did not significantly impact OS. Since only KPS was significant in univariate analyses, multivariate analysis was not performed.

## 4. Discussion

The previous literature on spinal SIB–SBRT is very limited. Prospective data are awaited, as Cellini et al. are currently evaluating SIB–SBRT with 3 × 7/10 Gy for pain control [29] and Sprave et al. are comparing different SBRT regimens including SIB strategies (10 × 3/4 Gy and 5 × 4/6 Gy) regarding local tumor control [30]. A retrospective analysis including 20 spinal metastases (comprising 71.1% of the whole study population) treated with 10 × 3/4 Gy reports a one-year local tumor control rate of 90% [31]. We observed excellent and durable local control rates of 98.6% and 96.4% at one and two years after SIB–SBRT in five fractions. 

Comparing our results to previously published local control rates after spinal SBRT without SIB, we observe at least comparable, if not superior, results; Shagal et al. prospectively detected a local control rate of approx. 97% at six months after SBRT with 2 × 12 Gy [6], while pooled local control rates of 91% at one year following SBRT with 1 × 24 Gy and 3 × 9 Gy [23], and 82% at a median follow-up of 45 months after 5 × 7 Gy or 10 × 4.85 Gy [22] were observed in cohorts that included both radiosensitive and radioresistant tumors. In a large retrospectively assessed series of 360 lesions treated with non-SIB SBRT with cumulative doses of 24 to 27 Gy in two to three fractions, local control was 81.1% and 70.6% at one and two years, respectively [21]. Interestingly, patients with prostate cancer were excluded in this analysis. We observed excellent and significantly superior local control for prostate cancer metastases without a single local recurrence in our cohort, and comparably good results after prostate-cancer-metastases-directed SBRT have been reported in other series [32]. This might explain, at least to a certain degree, the comparably low local control rates observed by Chen et al. [21], and suggests that patients with radioresistant tumors might further benefit from dose escalation, for example, utilizing an SIB regimen. 

Chen et al. demonstrated that adherence to the available consensus guidelines on target volume definition leads to improved local control [21]. We utilized a target volume definition strategy for PTV generally following these guidelines as well, providing further data indicating that guideline adherence yields excellent results in terms of local tumor control. 

We acknowledge that there was no uniform imaging follow-up strategy after SIB–SBRT, and in a few cases, only PSA follow-up was available. However, we reported detailed PSA dynamics following SIB–SBRT and provided information on anti-hormonal therapy as well, justifying the rationale for local control assumption in these cases. Given that most recurrences are expected to occur within one year after treatment [21], providing a median follow-up of 18 months for local failure and with median overall survival not being reached, we assume that it is unlikely that our observed control rates are relevantly confounded by death. 

We demonstrated that SIB–SBRT is a safe strategy for dose escalation. It was tolerated considerably well, with only one single treatment-related grade III toxicity (VCF requiring surgery). Estimated freedom from VCF was 96.1% and 90.4% at one and two years, which is low in comparison with previously published studies [24,25]. As the majority of VCF develops within a few months after spinal SBRT and providing a median local follow-up of 18 months, we assume that SIB–SBRT of spinal metastases leads to durably low VCF rates. However, the heterogenous follow-up imaging certainly limits our safety analyses regarding VCF; although most patients did receive structured imaging follow-up, the rate of asymptomatic VCF might be underestimated. Nevertheless, the observed 3% rate of patients requiring surgical intervention for SIB–SBRT-related VCF is indeed low [24], demonstrating the safety of this treatment concept. Osteoporosis was identified as significantly increasing the risk for VCF, and patients should accordingly be advised. The fact that known risk factors for VCF development such as SINS could not be confirmed in our analysis might be related to the small number of events. 

Interestingly, we observed VCF in cases with rather large treatment volumes, and detected a trend for higher VCF rates with larger PTV volumes (*p* = 0.08). It has been previously reported that it is more likely to observe post-SBRT VCF when larger proportions of the spine segment receive relevant doses, with D80% of 25 Gy and D50% of 28 Gy in three fractions corresponding to 10% VCF risk [33]. Utilizing an SIB regimen might be an effective yet safe strategy to avoid higher doses to a relevant proportion of the spine segment, while treating the macroscopic metastasis with sufficiently high doses. We also therefore strongly recommend adherence to consensus guidelines, avoiding overtreatment with unnecessary inclusion of relevant proportions of or even the whole vertebra. A median PFS of 10 months and fairly high overall survival rates of 91.5%, 80.2% and 64.8% at one, two and five years, respectively, demonstrate that local ablative therapy has the potential to favorably influence the course of disease in oligometastatic patients. SIB–SBRT may be used in selected patients with oligometastatic disease to improve prognosis.

## 5. Conclusions

To the best of our knowledge, we provide the first clinical data on SBRT to spinal metastases using an SIB dosing concept in a larger patient cohort. Our analyses encourage the use of hypofractionated SIB–SBRT as a safe and effective strategy for dose-escalated treatment for spinal metastases. Larger series and prospective data confirming the effectivity and safety of SIB–SBRT are warranted, and this treatment regimen might as well be utilized as a possible SBRT strategy for the treatment of spinal metastases in prospective clinical trials. Regarding survival outcomes, oligometastatic cancer patients with prostate cancer, secondary oligometastatic disease and good performance status might especially benefit from this treatment approach. SIB–SBRT may therefore be utilized in selected oligometastatic cancer patients to improve prognosis.

## Figures and Tables

**Figure 1 cancers-15-05813-f001:**
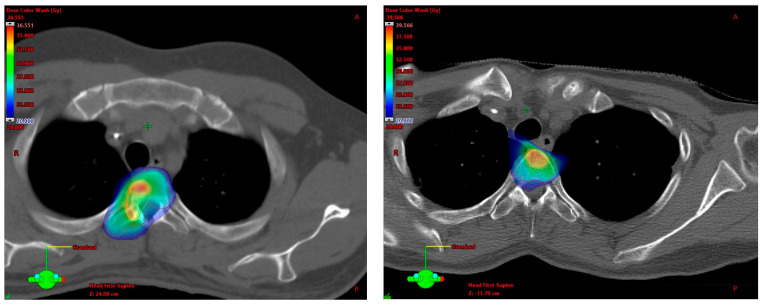
Exemplary cases for spinal SIB–SBRT treatment plans with prescription doses of 5 × 5/7 Gy, with PTV in red and SIB in orange. (**Left**): Rhabdomyosarcoma metastasis in the lateral body of T3 extending into the right pedicle; PTV includes vertebral body, pedicles, the right transverse process and lamina (with incomplete dose coverage of SIB due to adherence to dose constraints for myelon). (**Right**): Leiomyosarcoma metastasis centralized in the body of T3; PTV includes the vertebral body.

**Figure 2 cancers-15-05813-f002:**
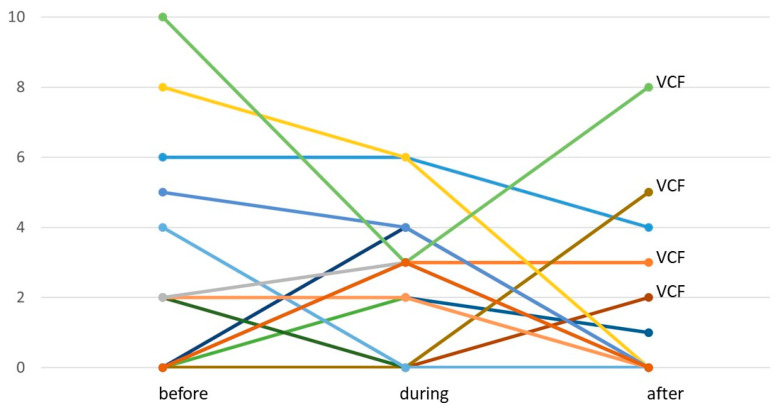
Pain dynamics of all patients reporting pain before, during and/or after SIB–SBRT (*n* = 19 patients with *n* = 20 lesions; yet, long-term follow-up (=after) was not available for *n* = 4 patients with *n* = 4 lesions). Each graph refers to a specific lesion.

**Figure 3 cancers-15-05813-f003:**
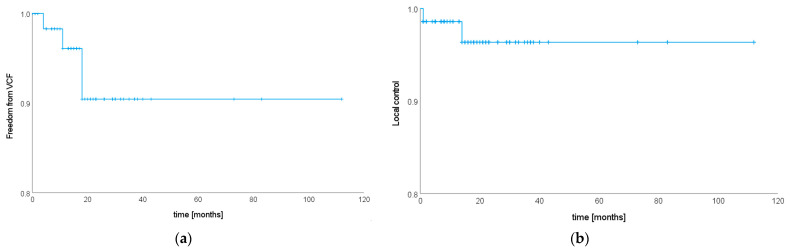
Kaplan–Meier curves showing (**a**) freedom from VCF (*n* = 64 lesions) and (**b**) local tumor control after SIB–SBRT (*n* = 71 lesions).

**Figure 4 cancers-15-05813-f004:**
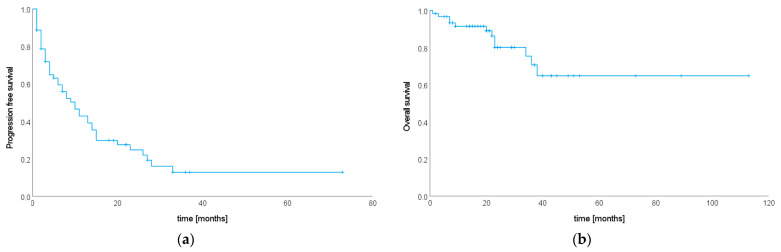
Kaplan–Meier curves showing (**a**) progression-free survival and (**b**) overall survival after SIB–SBRT in months (*n* = 62 patients).

**Figure 5 cancers-15-05813-f005:**
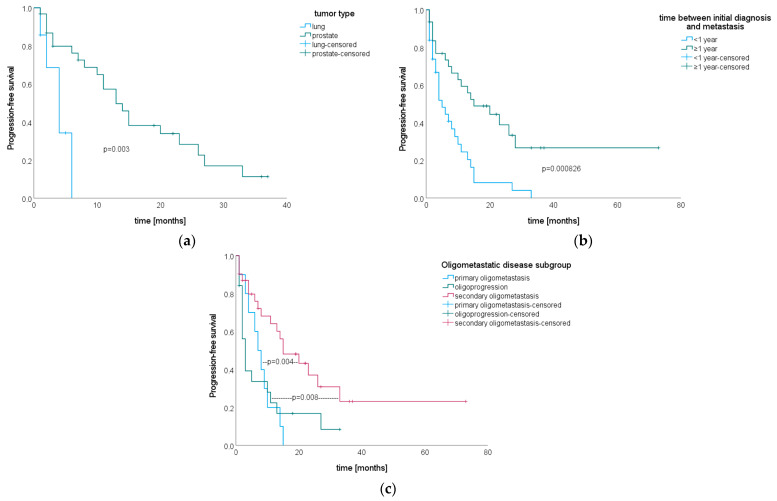
Kaplan–Meier curves for progression-free survival after SIB–SBRT stratified by (**a**) tumor type (*n* = 7 patients with lung cancer, *n* = 31 patients with prostate cancer), (**b**) time between initial diagnosis and development of metastases (*n* = 31 with <1 year, *n* = 31 patients with ≥1 year) and (**c**) subclassification of oligometastatic disease (*n* = 10 patients with primary oligometastasis, *n* = 19 patients with oligoprogression and *n* = 31 patients with secondary oligometastasis.

**Figure 6 cancers-15-05813-f006:**
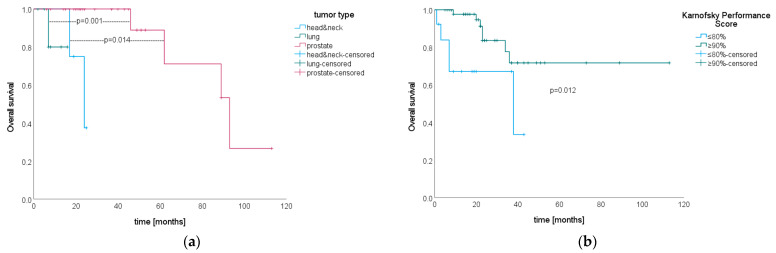
Kaplan–Meier curves for overall survival after SIB–SBRT stratified by (**a**) tumor type (*n* = 4 patients with head and neck cancer, *n* = 7 patients with lung cancer and *n* = 31 patients with prostate cancer) and (**b**) KPS (*n* = 13 patients with ≤80%, *n* = 48 patients with ≥90%).

**Table 1 cancers-15-05813-t001:** Patient characteristics for *n* = 62 patients with *n* = 71 SIB–SBRT lesions comprising the study population; ^+^ data refer to individual patients; * data refer to SIB–SBRT lesions; NA: not available.

Characteristic	*n*	%
Gender ^+^		
male	43	69.35
female	19	30.65
Age at primary diagnosis ^+^		
20–40	7	11.29
41–60	24	38.71
61–80	29	46.77
>80	2	3.23
median 60.5 (R 20–91)		
Age at SIB–SBRT ^+^		
30–40	2	3.23
41–60	22	35.48
61–80	33	53.23
>80	5	8.06
median 66.5 (R 30–91)		
Years from initial diagnosis to development of metastases		
≤1	31	50
>1	31	50
median 0.96 (R 0–27)		
Primary tumor ^+^		
prostate	31	50.00
breast	9	14.52
lung	7	11.29
head and neck cancer	4	6.45
melanoma	3	4.84
other	8	12.90
Karnofsky Performance Score ^+^		
90–100	48	77.42
70–80	8	12.90
50–60	5	8.06
NA	1	1.61
median 100 (60–100)		
Postmenopausal status ^+^		
male gender	43	69.35
premenopausal	1	1.61
postmenopausal	8	12.90
NA	10	16.13
Diagnosed osteoporosis ^+^		
yes	1	1.61
no	61	98.39
Oligometastatic subgroup ^+^		
primary oligometastasis	10	16.13
secondary oligometastasis	31	50.00
oligoprogression	19	30.65
oligopersistence	2	3.23
Number of active metastases present at SIB–SBRT ^+^		
1	39	62.90
2	15	24.19
3	8	12.90
Location of spinal metastasis *		
cervical spine	4	5.63
thoracic spine	34	47.89
lumbar spine	30	42.25
sacrum	3	4.23
Type of metastasis *		
osteolytic	17	23.94
osteoblastic	45	63.38
mixed	9	12.68
Baseline pain (VAS) *		
0	62	87.32
2	4	5.63
4–6	3	4.23
8–10	2	2.82
Spinal Instability Neoplastic Score (SINS) *		
1–6 (stable)	67	94.37
7–9 (potentially unstable)	4	5.63

## Data Availability

Available from the authors upon reasonable request.

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
