# Peer review of "SBRT of Spinal Metastases Using a Simultaneous Integrated Boost Concept in Oligometastatic Cancer Patients Is Safe and Effective"

_cancers, 2023, doi:10.3390/cancers15245813_

Round 1
Reviewer 1 Report
Comments and Suggestions for Authors
A single arm retrospective single centre study of a difference dose fractionation than the standard 27Gy in 3 fractions.
Well presented, but small case case series.
A larger Randomised controlled trial comparing to the 27Gy in 3 fractions is needed.
The research is looking at the safety and efficacy of a new SABR fractionation of oligometastatic prostate cancer. Spinal metastatic disease treated with a novel fractionation.
The topic is original and this is the first published trial using the new fractionation. This is useful to inform local guidelines and decision making.
This study has added significantly with an emphasis on tolerability and safety.
The specific improvements the authors should consider regarding the
methodology and further controls should be considered: This is a patient registry and requires an RCT comparing to the standard fractionation.
The conclusions are consistent with the evidence and proves the efficacy of the specific treatment.
references: Publication is well written and compares to all well known publications in the subject. The introduction highlights the issue and the discussion compares to current practice. The references are appropriate.
tables and figures: The data are well presented. The pain score figure shows the scale of benefit for symptoms.
Author Response
We thank the reviewer for the valuable feedback. Please find a point-by-point response below. Changes are highlighted in yellow in the manuscript.
Response:
We thank the reviewer for the positive feedback, and agree that a RCT comparing different fractionation schemes for spinal SBRT might be of additional value and could possibly lead to identifying a superior treatment scheme that could be recommended in clinical routine. We also believe that our encouraging results should be verified prospectively, as now pointed out in the conclusions section (see line 481). However, there exist several non-SIB fractionation schemes frequently used for spinal SBRT (such as 3 x 9 Gy, 2 x 12 Gy and 5 x 6 Gy), and local control rates might not only or necessarily depend on the fractionation, but also on the underlying tumor type, as already discussed (see line 430 following).
There are currently being conducted two RCTs comparing SIB- and non-SIB-SBRT fractionation schemes, one of which assessing local tumor control as primary endpoint[1]. Results are awaited. We already addressed this in the discussions section (see line 414 following).
Reference
1 Sprave T, Welte SE, Bruckner T, et al (2018) Intensity-modulated radiotherapy with integrated-boost in patients with bone metastasis of the spine: Study protocol for a randomized controlled trial. Trials 19:1–6. https://doi.org/10.1186/s13063-018-2452-7
Reviewer 2 Report
Comments and Suggestions for Authors
This study is a retrospective single-arm study to assess safety and effectivity of stereotactic body radiotherapy (SBRT) to spinal metastases. Authors retrospectively analyzed toxicity and survival outcomes (progression-free survival (PFS) and overall survival (OS)) and assessed potential influential factors using Kaplan-Meier estimator, log-rank test, and Cox regression models. Even though the approach looks fine in overall, I think that there also exist several methodological concerns.
1. This is a retrospective study and the study data includes distinct baseline characteristics (Table 1) like gender effect (69.35% vs 30.65%), age at SIB-SBRT+, number of metastases, etc. Please describe how authors handled preexisting covariates from the analyses in Methods section.
2. In Materials and Methods, please describe how you summarized the data (for both continuous and discrete data). For example, Frequencies (n) and percentages (%) were summarized for categorical data while medians and ranges were calculated for continuous variables. In addition, please describe what are your criteria or cut-points to include significant parameters into a subsequent multivariate analysis using Cox regression analyses.
3. These are other comments related to the Materials and Methods section. (1) Please define the survival time for progression-free survival (PFS) and overall survival (OS) separately because two outcomes are different. (2) This sentence (line 117) needs to be rewritten because log-rank tests are used to compare the two distributions of survival data.
4. In Results section, authors need to provide more detail and appropriate figures. In 3.4 and 3.4 of Results section, authors need to provide figures related to differences among PFS and OS distributions. For example, you can include several K-M curves on (1) PFS after SIB-SBRT by tumor type (p=0.11), (2) sub-329 classification of oligometastatic state (p=0.014), KPS ≥ 90% at SIB-SBRT (OS, p=0.012), prostate cancer tumor types with lung (p=0.014) and head and neck cancer (p=0.001), etc.
5. These are comments related to Table 1. (1) 1. Lines (130-131): What stands for “R” from 60.5 years (range, R 20-91)? Please describe it in Methods section. (2) In Table 1, you don’t need to “%” because you mentioned “%” in top row. For male, just “69.35”.
6. To authors: If I am wrong, please let me know. In Figure 3, do you think these K-M curves provide sufficient and necessary information for your study? Are they (freedom from VCF and local tumor control) informative with K-M analyses?
7. In Supplementary Materials, I clicked www.mdpi.com/xxx/s1 but no materials could be downloaded.
Supplementary Materials: The following supporting information can be downloaded at: 433 www.mdpi.com/xxx/s1, Table S1: Detailed information on target volume definition for SIB-SBRT 434 based on the localization of the metastasis within the vertebra at patient-level.
Author Response
We thank the reviewer for the valuable feedback. Please find a point-by-point response below. Changes are highlighted in yellow in the manuscript.
This study is a retrospective single-arm study to assess safety and effectivity of stereotactic body radiotherapy (SBRT) to spinal metastases. Authors retrospectively analyzed toxicity and survival outcomes (progression-free survival (PFS) and overall survival (OS)) and assessed potential influential factors using Kaplan-Meier estimator, log-rank test, and Cox regression models. Even though the approach looks fine in overall, I think that there also exist several methodological concerns.
- This is a retrospective study and the study data includes distinct baseline characteristics (Table 1) like gender effect (69.35% vs 30.65%), age at SIB-SBRT+, number of metastases, etc. Please describe how authors handled preexisting covariates from the analyses in Methods section.
Response:
Our study reports on a standardized treatment concept i.e., hypofractionated stereotactic radiotherapy to spinal metastases utilizing a simultaneous integrated boost, and does not compare two different treatment regimens. Therefore, analyses regarding the distribution of characteristics across different treatment groups are not needed.
As rightly pointed out by the reviewer, the study population consists of real-world data that is not homogenous and therefore baseline characteristics such as age or gender are not evenly distributed. We however performed significance analysis (log-rank test) for potentially relevant patient, tumor and SIB-SBRT-related characteristics with regard to PFS and OS (gender, performance status, age at initial diagnosis, at the development of metastases and at SIB-SBRT, number of metastases at SIB-SBRT, use of systemic therapy in the treatment concept for oligometastatic disease tumor type, time to development of metastases, subclassification of oligometastatic state, RT dose).
We thank the reviewer for this comment and added a detailed description of the covariates assessed in the methods section (see line 112 following). Furthermore, we have added the parameter time to development in Table 1 as it was missing in the initial manuscript (see results section, line 152 following).
- In Materials and Methods, please describe how you summarized the data (for both continuous and discrete data). For example, Frequencies (n) and percentages (%) were summarized for categorical data while medians and ranges were calculated for continuous variables. In addition, please describe what are your criteria or cut-points to include significant parameters into a subsequent multivariate analysis using Cox regression analyses.
Response:
Thanks for pointing out the need to report the strategies we utilized for summarizing and significance testing of the covariates, as we included both categorial and continuous variables. We added this information in the methods section (see line 112 following). A p-value <0.05 was utilized as cutoff for including parameters into a subsequent multivariate analysis. We added this information in the methods section (see line 135).
We have furthermore extended the univariate analyses for continuous variables (age at initial diagnosis, at the development of metastases, at SIB-SBRT and time from initial diagnosis to the development of metastases) performing additional Cox proportional hazards models (see methods section, line 128 following and results section, line 364 following for PFS and line 397 following for OS).
- These are other comments related to the Materials and Methods section. (1) Please define the survival time for progression-free survival (PFS) and overall survival (OS) separately because two outcomes are different. (2) This sentence (line 117) needs to be rewritten because log-rank tests are used to compare the two distributions of survival data.
Response:
We thank the reviewer to highlight the need to provide the information on PFS and OS definition and accordingly provided additional information on patient censoring regarding PFS and OS calculation (see methods section, line 114 following). The sentence “Kaplan–Meier estimator was used for assessment of VCF rates, local tumor control and survival analyses, and univariate analyses were performed using log-rank test.”, now “Parameters possibly influencing VCF, LC, PFS and OS were tested for significance using log-rank test.” was rewritten according to the valuable advice of the reviewer (see methods section, line 112 following)
- In Results section, authors need to provide more detail and appropriate figures. In 3.4 and 3.4 of Results section, authors need to provide figures related to differences among PFS and OS distributions. For example, you can include several K-M curves on (1) PFS after SIB-SBRT by tumor type (p=0.11), (2) sub-329 classification of oligometastatic state (p=0.014), KPS ≥ 90% at SIB-SBRT (OS, p=0.012), prostate cancer tumor types with lung (p=0.014) and head and neck cancer (p=0.001), etc.
Response:
In accordance with the reviewer, we believe that the results can be further illustrated by adding Kaplan-Meier curves for covariates showing significant survival differences and thank the reviewer for this important advice. We accordingly added the respective figures (see results section, line 379 following for PFS and line 407 following for OS).
- These are comments related to Table 1. (1) 1. Lines (130-131): What stands for “R” from 60.5 years (range, R 20-91)? Please describe it in Methods section. (2) In Table 1, you don’t need to “%” because you mentioned “%” in top row. For male, just “69.35”.
Response:
(1) We apologize for the lack of clarity that “R” was used as abbreviation for “range”. We revised the section and believe that the abbreviation now should be attributed unambiguously to “range” (see results section, line 149).
(2) Thanks, it has been accordingly revised (see results section, line 152 following).
- To authors: If I am wrong, please let me know. In Figure 3, do you think these K-M curves provide sufficient and necessary information for your study? Are they (freedom from VCF and local tumor control) informative with K-M analyses?
Response:
We thank the reviewer for this comment. We do believe that it is a commonly applied method to illustrate local tumor control with Kaplan-Meier curves and are of the opinion that Kaplan-Meier curves can as well be used to illustrate freedom from VCF. Kaplan-Meier curves visualize the censored patients which is important for assessing the level of validity of the reported local tumor control rate and rate of freedom from VCF. We now provide further data on censored patients as well (see results section, line 269 following for VCF and line 309 following for local tumor control), reporting the proportion of patients lost to imaging follow up and patients who had died. Due to the low proportion censored patients due to death (7.81% and 7.04% at one year after SIB-SBRT for VCF and local control), further competing risk analyses (death as a competing risk to local recurrence or VCF) were not performed. We also addressed this in the discussion. For better readability, we adjusted the scale on the y-axes (ranging from 0.8 to 1 now instead of from 0 to 1, see results section, line 328 following).
- In Supplementary Materials, I clicked www.mdpi.com/xxx/s1 but no materials could be downloaded.
Supplementary Materials: The following supporting information can be downloaded at: 433 www.mdpi.com/xxx/s1, Table S1: Detailed information on target volume definition for SIB-SBRT 434 based on the localization of the metastasis within the vertebra at patient-level.
Response:
Thank you for correctly pointing out this inaccuracy. We have mistakenly included the table in supplements (and therefore added this section as requested in Cancers’ manuscript template), although we already had included the same table it in appendix. After consultation with the assistant editor, we deleted the Supplementary Materials statement and provide the table only in appendix (referred to as Table A1 now).
Round 2
Reviewer 2 Report
Comments and Suggestions for Authors
I reviewed authors' revised version and their answers on my critiques, and they really replied well for my comments. Now I do not have any further issues on this revised version.